# Growing Jatropha (*Jatropha curcas* L.) as a Potential Second-Generation Biodiesel Feedstock

**Dhurba Neupane** [1,*], **Dwarika Bhattarai** [2], **Zeeshan Ahmed** [3], **Bhupendra Das** [4], **Sharad Pandey** [5], **Juan K. Q. Solomon** [6], **Ruijun Qin** [7] **and Pramila Adhikari** [8,*]

1  Department of Biochemistry & Molecular Biology, University of Nevada, Reno, NV 89557, USA
2  Department of Agronomy, Horticulture, and Plant Science, South Dakota State University, Brookings, SD 57007, USA; Dwarika.Bhattarai@sdstate.edu
3  Cele National Station of Observation and Research for Desert-Grassland Ecosystem, Xinjiang Institute of Ecology and Geography, Chinese Academy of Sciences, Urumqi 830011, China; zeeshanagronomist@yahoo.com
4  Nepal Energy and Environment Development Services (NEEDS), Kathmandu 44600, Nepal; bhupenids@gmail.com
5  National Centre for Fruit Development, Kathmandu 44600, Nepal; pandey.sharad@gmail.com
6  Department of Agriculture, Veterinary & Rangeland Sciences, University of Nevada, 1664 N. Virginia Street, Reno, NV 89557, USA; juansolomon@unr.edu
7  Hermiston Agricultural Research & Extension Center, Oregon State University, 2121 South 1st Street, Hermiston, OR 97838, USA; ruijun.qin@oregonstate.edu
8  Department of Social Work, Tribhuvan University, Kathmandu 44600, Nepal
*  Correspondence: dneupane@unr.edu (D.N.); adhpramila@yahoo.com (P.A.)

**Abstract:** Dwindling supplies of fossil fuels and their deleterious impacts on human health and the global environment have intensified the search for substitute energy sources. Biodiesel has been identified as a promising renewable energy substitute for diesel fuel due to several comparable and sustainable properties. However, approximately 95% of biodiesel is derived from edible oil crops, threatening the current food supplies. Therefore, the biodiesel production potential from inexpensive, non-edible, and non-conventional bioenergy crops, such as Jatropha (*Jatropha curcas* L.), has attracted the attention of many researchers, policymakers, and industries globally. Jatropha is considered to be the second-generation biofuel feedstocks for biodiesel production. However, sustainable biodiesel generation from *J. curcas* oil has not yet been attained, owing to different socio-economic, ecological, and technical factors. This study aimed to synthesize the information from the existing literature on the present status and to identify the knowledge gaps for future research on Jatropha by providing comprehensive information regarding its origin and distribution, morphology, phenology, and reproduction, genetic diversity, its productivity, oil content, and fatty acid composition, the methodology used for extracting biodiesel, and agronomic, economic, and environmental aspects of biodiesel production. The germplasm screening of *J. curcas* and the exploration of its adaptability and agronomic potential across diverse climates are highly desired to promote this crop as an alternative biofuel crop, particularly in arid and semi-arid regions. Moreover, future research should focus on developing, optimizing, and modernizing the technologies involving seed collection, the processing of seeds, oil extraction, and the production of biodiesel.

**Keywords:** *Jatropha curcas* L.; non-edible feedstock; biodiesel production; arid regions; semi-arid regions

## 1. Introduction

The global demand for biofuel production has fueled exponentially due to the global concern toward energy security and environmental sustainability in terms of net lifecycle greenhouse gas (GHG) emissions associated with the use of fossil fuels, which are expected to reach approximately 37 gigatons (Gt) in 2035 from 31 Gt in 2011 [1–3]. Worldwide, renewable energy accounts for 13% of the total energy consumption, in which bioenergy

accounts for ~10% [2]. Biofuels (e.g., biodiesel and bioethanol) for transport represent the major proportion of bioenergy production, which is approximately 50% of oil consumption worldwide, with approximately 25% of global energy-related carbon dioxide ($CO_2$) emissions [4]. Biodiesel consists of monoalkyl esters of fatty acid produced by the transesterification of vegetable oils or animal fats [5,6]. Some of the advantages of biodiesel are its renewability, portability, safety to use in all kinds of diesel engines, its good performance and engine durability, which is similar to petroleum diesel fuel, the fact that it's nontoxic, nonflammable, has lower tailpipe emissions, a low sulfur and aromatic content, higher cetane number, and higher biodegradability [5]. The favorable alternatives for use as a potential biodiesel are vegetable oils that are derived from palm oil, soybean oil, sunflower oil, canola oil, and rapeseed oil, because they are renewable, environmentally friendly, and easily available even in rural areas, as well as low-cost feedstock containing fatty acids, such as non-edible oils, animal fats, by-products from refining vegetable oils, and waste food oils. Further, the use of vegetable-based products helps to promote rural economic development, particularly in developing countries, because the farmers would directly benefit from the increased demand for vegetable oils [1]. This will help to decrease the dependency on fossil fuel imports, improve their economic conditions, and can create new employment opportunities, especially in the agriculture area [7]. The use of biodiesel as a fuel has increased substantially in recent years; however, the feedstock cost accounts for a greater percentage of the direct biodiesel production cost, including the capital cost and return [8]. Moreover, the use of first-generation biofuel crops, such as corn, sugarcane, wheat, sugar beet, cassava, rapeseed, soybean, and oil palm, which contribute to more than two-thirds of the bioenergy demand of the world, pose a growing concern over the competition for land, food, and water resources to produce energy [2,9,10]. Some of the second-generation biofuel feedstocks from non-edible sources include camelina (*Camelina sativa*) [11–15], *Brassica carinata* (*Brassica carinata*) [16], Jatropha (*Jatropha curcas* L), karanja (*Pongamia pinnata*), tobacco seed (*Nicotiana tabacum* L.), mahua (*Madhuca indica*), neem (*Azadirachta indica*), the rubber seed tree (*Hevea brasiliensis*) and microalgae [5]. In particular, camelina, karanja, and Jatropha have received significant attention due to both their tolerance to growing in poor soils and marginalized lands and their low moisture demand [9].

Jatropha (*J. curcas* L.), also known as the physic nut, is a large shrub or small tree that belongs to the genus Euphorbiaceae, which produces oil-containing seeds. This species is native to North and Central America, and is now widespread all over the tropical and subtropical regions of the world, such as Africa, India, Southeast Asia, and China [17–20]. *Jatropha* species are well adapted to extreme drought and soil conditions, and are thus able to grow well in semi-arid and arid regions, with high temperatures and a low soil moisture [19,20]. There has been a recent interest in growing *J. curcas* as a potential biofuel (biodiesel) plant in non-oil production countries due to its adaptability to grow in moderately sodic and saline soil and marginal land with minimum water and energy requirements [9,20].

The increased interest in growing *J. curcas* has been supported by the evidence of a significant increase in peer-reviewed publications from the database comprising the word "Jatropha and biodiesel" using the Google search engines, such as Google Scholar (22,500), ScienceDirect (6276), and Web of Science (3434), published from 2000–2021 (reported on 10 March 2021). The evidence of many publications and data reported on *J. curcas* reveal the huge potential of this crop in climate change scenarios. Although intensive studies on *J. curcas* as a biofuel were reported with encouraging results, no reviews on the topic describing all of the aspects of *J. curcas*, such as its origin and distribution, morphology, phenology and reproduction, genetic diversity, the productivity, fatty acid composition, composition, and characteristics of Jatropha oil, the methodology used for extracting biodiesel, and the agronomic, economic, and environmental aspects of biodiesel production, were found in the literature. Thus, this study aimed to synthesize the information from the existing literature on the present status and to identify the gaps in knowledge and areas for its improvement in future research.

## 2. Origin, Distribution and Exploitation of *J. curcas*

The origin of *J. curcas* is still contradictory. The earliest remains of the genus *Jatropha* were found in the geological formations of Peru, South America approximately 66 million to 3.0 million years ago (i.e., early Tertiary age), and the closest relatives of this genus were found in wet tropical regions of Central and northern South America during the late Miocene, approximately 23 million to 5.0 million years ago [21]. In addition, it was alienated from the African continent in the Albian approximately 100 million years ago [22], where *Jatropha* was introduced by the Portuguese in recent years [23]. If we track this continental formation and rely on Berry's documentation, there is no doubt when stating that the origin of *Jatropha* is South America. The most primitive species, *J. curcas*, and its relatives were found in Mexico, suggesting this region as the center of origin [24,25]. Molecular biomarkers of *Jatropha* from several countries have shown similarities in the molecular profile with the species found in Mexico [26]. In addition, Li et al. [27] support Mexico as the center of origin of *Jatropha* based on their research on genetic tracing. According to USDA-ARS [28], *J. curcas* is native to Mexico (Chiapas) and Southern America, including Belize, Costa Rica, Guatemala, Honduras, Nicaragua, El Salvador, Brazil, Bolivia, Peru, Argentina, and Paraguay.

As mentioned earlier [23], the book 'Physic nut, *J. curcas* L.' claimed that Portuguese seafarers transported the seeds of *J. curcas* to Cape Verde, Africa and Asia in the 1800s. Being a tropical crop, it is widely distributed along the equatorial line and the tropical countries across the world. The map (Figure 1) displays the distribution of Jatropha crops around the world; their native region is Central and South America, and they were introduced to the majority of areas of Africa, South Asia, and Australia.

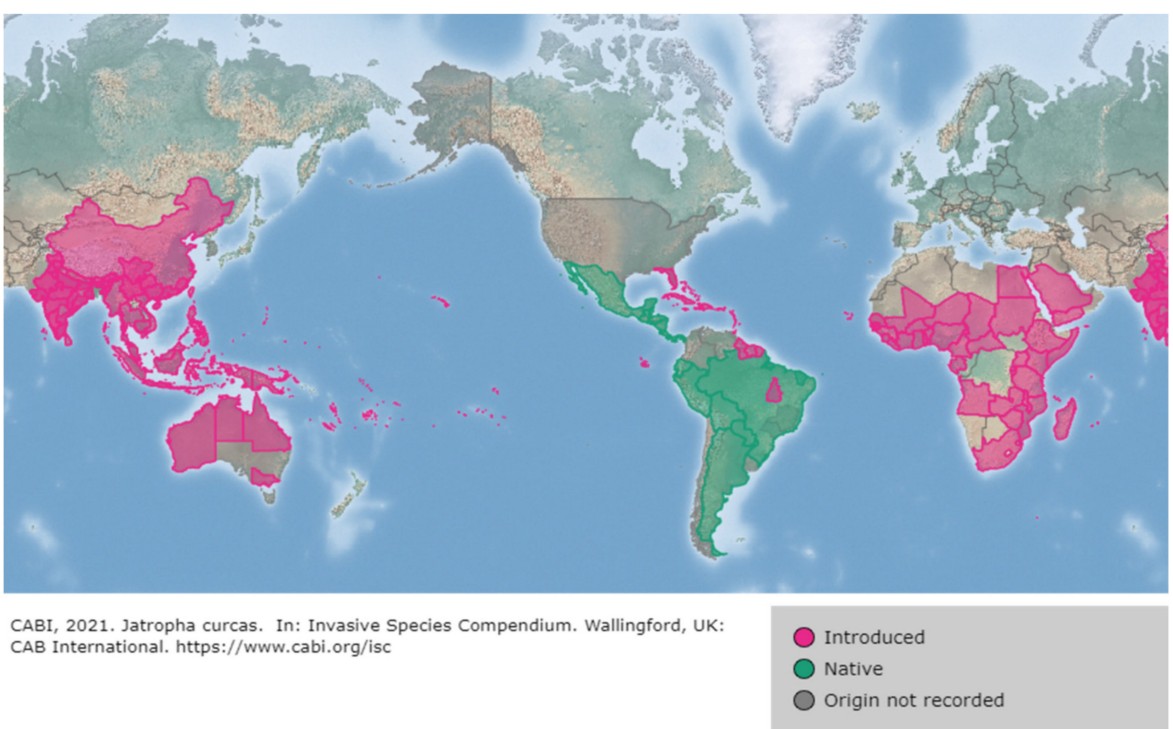

CABI, 2021. Jatropha curcas. In: Invasive Species Compendium. Wallingford, UK: CAB International. https://www.cabi.org/isc

● Introduced
● Native
● Origin not recorded

**Figure 1.** Map showing the distribution of *J. curcas* across the world. Source: CABI, 2021. https://www.cabi.org/isc (accessed on 25 June 2021).

*J. curcas* has various applications, including erosion control, establishing live fences, enhancing soil, livestock feedstock, and the production of biofuels [9], along with its medicinal uses; however, it is non-edible because of toxins, such as phorbol esters [29]. The leaves of *J. curcas* are composed of flavonoids, sterol, alkaloids, and alcohol [30], which are responsible for various medicinal values [31], along with anti-inflammatory

activities [32]. Further, Singh et al. [33] reported that the leaves can be used to treat vaginal bleeding. Various parts of *J. curcas* contain chemicals, such as sapogenin, which are used to produce soap and biocides (i.e., insecticides, fungicide, molluscicide, and nematicide) [34]. Similarly, the direct use of Jatropha oil without any reformations can be used in older engines and motors that operate with current technologies; for example, pumps and generators [35]. Moreover, the seed oil is composed of oleic, linoleic, and palmitic acids, which make the oil feasible for biofuel production [36]. Various uses of different parts of *J. curcas* are summarized in Table 1.

**Table 1.** Table showing the exploitation of *J. curcas.*

| Plant Parts | Uses | References |
|---|---|---|
| Leaves | Medicinal uses | [23,31] |
| | Anti-inflammatory | [32] |
| | Silk farming | [37] |
| Latex | Coagulant and anticoagulant | [38] |
| | Antimicrobial toothpaste | [39] |
| | Wound healing | [40] |
| Seed oil | Biofuel | [41] |
| | Medicinal uses | [42] |
| | Soap production | [43] |
| Seed cake | Fertilizer | [44] |
| | Biogas | [45] |
| | Fodder (non-toxic) | [23] |
| Roots | Anti-microbial | [46] |
| | Medicinal uses | [31,47] |

## 3. Morphology and Phenology

*J. curcas* is a 2–5 m tall shrub that produces yellowish/milky, pungent, and astringent latex. Leaves are deciduous, and alternate between apically crowded, ovate, acute to acuminate, and basally cordate, with the dimensions of 6–40 cm L × 6–35 cm W. The plant is monoecious, and sometimes unisexual flowers are present at the terminal inflorescence. The flowers are yellowish, bell-shaped, and in greenish cymes. The inflorescence appears in the rainy season and consists of 80–90% male flowers at the subordinate positions, whereas female flowers (10–20%) are at the apices of the main stem and the branches of the inflorescence [23,48].

According to a *J. curcas* flower phenology study by A Hartana [49], flower initiation started at 0–3 days after initiation (DAI), small bud formation at 3–7 DAI, large bud formation at 7–20 DAI, and flower blooming at 20–39 DAI. The fruit contains an average of three seeds, which take approximately 60 days to ripen and form yellow and dark brown colors at the time of harvesting. It has a tap root system with four lateral roots.

Jatropha can be found in pastures, along roadsides, in wastelands, and in open woodlands of tropical and subtropical regions, at an altitude ranging from 200 to 800 m above sea level. It can tolerate heat and drought stress [50], temperature ranges between 18 to 28.5 °C, and an annual mean precipitation of 48 to 238 cm.

## 4. Genetic Diversity

The genetic variation of tree species (i.e., among and within populations) plays a crucial role, not only for the long-term evolutionary potential and survival of the species in terms of their ability to respond to environmental changes, such as the occurrence of new pests and pathogens, but it is also vital for advancement in breeding and domestication [51]. The results from molecular marker studies revealed that temperate and tropical trees maintain a high diversity within populations, due to their life-history traits, such as outcrossing and efficient gene dispersal [52]. Anthropogenic activities caused a reduction in genetic diversity; for example, the logging or destruction of trees, which reduce

populations to critical sizes, can cause genetic erosion along with a reduced gene flow, which may lead to elevated breeding, an increased divergence between populations, and genetic bottlenecks [53].

*J. curcas* is a cross-pollinated plant, which results in a high degree of variation in many characters and provides a plethora of opportunities for the breeders to perform the screening and selection of seed sources for the desired traits [54,55]. The studies using microsatellite markers showed a very low genetic diversity in African and Asian landraces, and Mexica populations from the regions of Veracruz, Puebla, and Morelos (mostly monomorphic). However, the populations from Chiapas were polymorphic and were expected to have a heterozygosity between 0.34 and 0.54 [51]. The differences in genetic diversity might be attributed to the differential adoption, selection criteria, selection procedure, and environment [56]. Multiple types of molecular marker systems, such as RAPD (random amplification of polymorphic DNA; genetic fingerprinting technique), AFLP (amplified fragment length polymorphism), SSR (simple sequence repeat), and ISSR (inter simple sequence repeat), were used to evaluate Jatropha germplasm. Previous studies showed modest levels of genetic diversity in Indian accessions concerning the seed yield and oil content by 400 RAPD [54,57], 42% molecular polymorphism and 100 ISSR (inter simple sequence repeat: genetic fingerprinting technique, 33.5% molecular polymorphism) primers [58]. Studies also reported medium to high levels of polymorphism in Indian Jatropha germplasm by RAPD (42.0–80.2%), AFLP (88.0%), and ISSR (33.5%) markers. Mexican Jatropha varieties revealed a small portion of RAPD (15.09%), and AFLP (16.49%), and a higher portion of the SSR (58.33%) marker. A high level of polymorphism (97%) using the ISSR marker was found in a natural population of Chinese Jatropha [59]. These results suggested low to medium to high levels of polymorphism were detected in Jatropha globally and regionally. Similarly, observations made among landraces collected from the Brazilian states using genetic marker systems reported a very high genetic uniformity [60,61]. Murty et al. [62] evaluated 19 Jatropha germplasm and reported very high levels of polymorphism using ISSR (90%), RAPD (96%), and DAMD (direct amplification of minisatellite DNA marker) (91%). Mavuso et al. [63] evaluated the genetic diversity of 78 Jatropha accessions cultivated in Taiwan using the ISSR marker and found 31.23% of genetic variability among populations, but 77% within populations, suggesting that there was a low variation in Taiwan Jatropha accessions. The genetic diversity of 50 *J. curcas* germplasm from Costa Rica was also evaluated using nuclear ribosomal DNA internal transcribed spacers (nrDNA-ITS), expressed sequence tags-SSR (EST-SSR), and genomic simple sequence repeats (G-SSR) markers [64]. Further, comparative studies showed a very high genetic uniformity, even among accessions from different continents, whereas the genetic variability was observed in accessions from the Mexican states [58,65]. A high degree of homozygosity plus a high genetic uniformity was reported among 907 accessions collected worldwide [65]. Similar results of high genetic uniformity and homozygosity of *J. curcas* accessions grown in several countries of South America, Africa, and Asia further supported the idea of the Mexico–Central America region as the center of origin of the Jatropha species [66].

## 5. Cultivation Practices of Jatropha

Various factors play a crucial role in *J. curcas* cultivation, which include the propagation method, soil quality, and water and nutrient level.

### 5.1. Propagation

*J. curcas* is mainly propagated asexually by cutting (a way of separating a branch from an existing tree and implanted) and sexually via direct seeding/seedling transplantation from a nursery to the field [67]. Cuttings are mainly prepared from one-year-old terminal branches of 25–30 cm. To ensure rooting in cuttings, they are pre-treated with indole-3-butyric acid (IBA) [68]. Sexually, seed formation occurs via self- or cross-pollination; apomixis can be performed for breeding purposes [69]. Seed propagation leads to a lot

of genetic variabilities in terms of growth, biomass, seed yield, and oil content. Before sowing, seeds are soaked in water for 24 h, and germination occurs in 5–10 days at 27–30 °C in saturated humid conditions. Inoculating cuttings with mycorrhizal fungi at the nursery provides plant–fungal symbiosis in field conditions [55]. Cuttings or seedlings should be grown in nurseries for 2–6 months before transplanting them in the field at the beginning of the wet season [48]. Although propagation by cutting saves the seed for oil production, plants grown by this method display a lower resistance to drought and disease because the new plant does not produce strong tap roots compared to the seeding method. Nonetheless, though seeding and seedling transplantations are unpredictable methods and lack a uniform yield, they are still considered the highest yielding methods, and are recommended for large-scale commercial production [9]. Micropropagation (or in vitro propagation) by seeds is an alternative to conventional methods for vegetative propagation and has received much attention in recent years due to its increasing multiplication rates and the fact that it produces plant materials free of viruses and other pathogens. It is the technical link in the generation of transgenic plants and somatically bred plants via tissue culture [70].

### 5.2. Soil Requirement

Jatropha can grow well in a wide range of soil types, ranging from arid, saline, and wastelands, except for waterlogged lands. It can thrive well on the poorest stony soil and in the crevices of rocks as well [55]. Clay and sandy soils were found to limit root growth and development [71]. Therefore, for the optimum growth of the plant, sandy loam or clay loam with good aeration and drainage, and a pH of 5.2–8.5, is most preferable. However, the salinity stress reduces the Jatropha growth and yield [72].

### 5.3. Irrigation and Fertilizer Management

*J. curcas* thrives well in various climatic regions with rainfall ranging from 250–1200 mm. Jatropha plantation is mainly found on land with an annual rainfall of 600 mm and a temperature ranging between 20 and 27 °C. However, an irrigation or rainfall amount of 900–1200 mm year$^{-1}$ is conducive for its optimum production [72]. Jatropha plantation may be successful in dry regions with an annual rainfall of 500–600 mm. Nonetheless, treated sewage effluent (TSE) can be a sustainable irrigation approach for obtaining a good yield of the crop, because TSE contains significant amounts of nitrogen (N), phosphorus (P), and potassium (K) [9].

Jatropha is sought to be the crop with minimal input requirements, such as fertilizers. However, an application of fertilizers is vital for its optimum growth and long-term seed yield because the continuous harvesting of seeds removes nutrients from the soil [73]. Optimum fertilizers rates for *J. curcas* reported from the previous literature are summarized in Table 2. Like other crops, NPK are typical nutrients for the growth of Jatropha. N is always vital for plant growth, whereas P is crucial for reproduction and seed formation, and K is vital for the plant to control osmotic processes [74]. Additionally, sulfur (S) also plays a significant role in increasing the oil content of seeds [75]. The most applied fertilizers for *J. curcas* cultivation are urea, diammonium phosphate (DAP), muriate of potash (MoP), and single superphosphate (SSP) [75]. It is critical to evaluate the soil properties, such as soil type, pH, organic matter, and NPK content, before designing the fertilizer plan.

**Table 2.** Different *J. curcas* fertilization treatments from the literature. This information is adapted from Alherbawi et al. [9].

| N:P: K Treatment (kg ha$^{-1}$ Year$^{-1}$) | Other Additives (kg ha$^{-1}$ Year$^{-1}$) |
| --- | --- |
| 68:69:75 | Sulphur (12.5) |
| 46:46:46 | - |
| 46:48:24 | - |
| 60:30:0 | - |
| 37:37:37 | cow dung (25) |

### 5.4. Insect, Pest, and Disease Management

*J. curcas* L. is resistant to herbivores; however, a diversity of arthropods has been found to be associated with it. Some of them are beneficial pollinators or entomophagous insects, and some are harmful or phytophagous [76]. A previous study displayed that those 15 hemipteran herbivores have been found in Nicaragua, many arthropod herbivores are present in Asia, and that two of them, namely *Scutellera nobilis* Fabr. (Heteroptera) and *Pempelia morosalis* Saalmuller (Lepidoptera), are the serious pests of *J. curcas* in India. Insect herbivores are also reported in Kenya, and flea beetles (*Aphothona* spp.; Coleoptera) caused serious threats in physic nut plantations in Africa [77].

A mosaic virus disease infestation has been found in a Jatropha plantation in India. This virus is transferred by the vector *Bemisia tobaci* Aleyrodidae. The disease is endemic to Jatropha and is not transmitted to any other plant. *Scutellera perplexa*, which is the insect predator, completed its life cycle on *J. curcas* and caused severe damage to developing fruit by sucking sap and consequently damaging premature fruit, leading to a substantial reduction in fruit, seed size, and yield [55].

Jatropha seeds contain phorbol ester, which is toxic to animals and humans and possesses insecticidal, molluscicidal, and fungicidal properties. Studies also revealed that Jatropha seeds are anthelmintic and are ground with palm oil to use as rat poison. In Ghana, leaves are used to fumigate houses against bed bugs [72]. Extracts obtained from different parts of *J. curcas* L., such as leaves, stems, roots, and seeds, display diverse properties to control pests and diseases of different plants. For example, an application of aqueous leaf extract can control *Ceratitis capitata* (fruit fly) larvae and insect pests (*Sitophilus zeamais* and *Rhyzopertha dominica*) in stored grain [78]. Ether extract demonstrates antibiotic properties against *Escherichia coli* and S*taphylococcus aureus*. Methanol extracts, which contain biodegradable toxins, are tested to control water snails in Germany. Oil and aqueous extracts are applied to control cotton bull worms, the pest of potatoes, pulses, and corn [72]. Similarly, aqueous leaf extract controlled *Ceratitis capitata* (fruit fly) larvae and insect pests (*Sitophilus zeamais* and *Rhyzopertha dominica*) in stored grain [78]. Leaf extract was found to be effective in controlling the larval states of *Spodoptera litura*, with a mortality rate of 60%. Further, methanol extract from seed oil controls 100% of the population of adult termites (*Odontotermes obesus*) [79]. Aqueous leaf extract from Jatropha can control *Colletotrichum musae*, the agent of anthracnose in bananas. In more recent studies, ethanolic extract from the *J. curcas* leaf reduced the germination of *Hemileia vastatrix* completely and inhibited the mycelial growth of *Cercospora coffeicola*, by 20% [78].

## 6. Biodiesel and Its Properties

Biodiesel is commonly known as fatty acid methyl esters (FAME), and it is produced by mixing methanol with vegetable oil, animal fat, or any other triacylglycerol-carrying material. Variation in feedstocks significantly changes the value of the characteristics of FAME. Various properties are directly associated with the composition of FAME, such as the cetane number, calorific value, viscosity, pour point, cloud point, iodine number, specific gravity, flashpoint, and cold filter plugging point (Table 3). Along with its inherent attributes, handling and manufacturing processes also affect the FAME properties of biodiesel fuel. These properties include the water content, ash content, acid number, cold soak filtration, sediment, metals, and methanol content [80,81]. The main physicochemical properties of biofuels generated from various feedstocks are discussed below.

**Table 3.** Fuel properties of biodiesel derived from different generation oil feedstock. This information is adapted from Singh et al. [81].

| Feedstocks | Density at 15 °C (kg/m³) | Heating Value (MJ/kg) | Cloud Point (°C) | Flashpoint (°C) | Pour Point (°C) | Viscosity at 40 °C (mm²/s) | Cetane Number | Iodine Number | Sulfur Content (wt.%) | Add Value mg/g |
|---|---|---|---|---|---|---|---|---|---|---|
| Animal fat | 875 | 36.73 | - | - | - | 4.25 | 63.88 | 83.02 | - | 0.38 |
| Ankistrodesmus | 869 | 40.72 | 7 | 144 | −6 | 4.19 | - | - | - | - |
| Babassu | 872 | 31.8 | 4 | 117 | - | 4.2 | 63.25 | - | - | 0.425 |
| Beef tallow | 832 | 40.23 | - | 152–171 | 15 | 4.89 | 60.36 | 44.4 | - | 0.2 |
| Bitter almond | 884 | - | 4.5 | 169 | −6 | 4.6 | 45.18 | 117.29 | - | 0.27 |
| Camelina | 885 | 45.2 | 2.5 | 150 | −6.3 | 4.11 | 48.91 | 146.5 | 3 ppm | 0.2 |
| Camelus dromedaries | 871 | 39.52 | 12.7 | 158 | 15.5 | 3.39 | 58.7 | 65.3 | 0.031 | - |
| Canola | 878 | 35.74 L | −3.25 | 172.36 | −8 | 4.42 | 54 | 113.6 | 2 ppm | 0.49 |
| Castor | 922 | 38.09 | −11.16 | 178.56 | −20 | 17.14 | 37.55 | 85.53 | 1.3 | 0.148 |
| Chicken fat | 883 | 40.17 | −7 | 172 | - | 4.98 | 48 | - | 23.45 | 0.22 |
| Chlorella variabilis | 867 | 38.78 | - | 157 | - | 4.875 | 58.6 | - | 0 | 0 |
| Coconut | 867 | 35.2 L, 38.2 H | −1.6 | 113.83 | −8.3 | 3.2 | 64.65 | - | 3 ppm | 0.18 |
| Cottonseed | 887 | 39.75 | 1.7 | 210 | −12.5 | 4.19 | 48.1 | 120 | - | 0.5 |
| Crambe abyssinica | 872 | 39.56 | - | 136 | - | 6 | - | - | - | - |
| Fish oil | 881 | 40.54 | - | 177 | - | 4.45 | 47 | - | - | - |
| Fish oil | 885 | 40.05 | - | 114 | - | 4.74 | 52.6 | - | - | - |
| Fleshing oil | 907 | 39.61 | - | - | - | - | - | 52 | >990 ppm | - |
| Groundnut | 920 | 39.8 | 8 | 132 | 3 | 4.4 | 59.85 | 71.8 | 1.315 ppm | - |
| Hazelnut | 896 | 39.58 | −7.65 | 172.7 | −6 | 4.81 | 62.95 | 109 | 7 ppm | 0.351 |
| *J. curcas* L. | 865 | 40.79 | 5.66 | 175.5 | 6 | 4.25 | 55.43 | 95.75 | 0.008 | 0.24 |
| Jojoba | 866 | 44.77 | - | 80.5 | - | 2.2–19.2 | 63.5 | 48.97 | 0.3 | 0.8 |
| Karanja | 889 | 36.56 | 13.3 | 157.4 | 6.4 | 4.79 | 56.55 | 89 | 0.003 | - |
| Kusum | 875 | - | - | 152 | −2 | 5.34 | - | 37.59 | <0.005 | 0.435 |
| Lard | 877 | 36.5 | - | 143.5 | 7 | 4.84 | - | 66–77 | - | 0.12 |
| Linseed | 852 | 37.45 | 2.43 | 241 | −9.6 | 3.95 | 34.6 | 178 | 0.002 | 0.335 |
| Mahua | 895 | 36.9 L, 39.4 H | 4.33 | 161 | −6.8 | 4.77 | 55 | 74.2 | - | 0.41 |
| Michelia champaca | 870 | 39.51 | - | 158 | - | 5.11 | 50.28 | 104 | - | 0.44 |
| Mustard | 879 | 40.4 | 16 | 169.16 | −18 | 5.53 | 56 | 128 | <1 | 0.2 |
| Neem | 886 | 39.84 | 114.5 | 144.75 | 7 | 6.09 | 51.26 | 46.84 | 473 ppm | - |
| Neem seed pyrolysis oil | 982 | 20.8 | - | 55 | - | 9.38 | - | - | - | - |
| Olive pomace | 894 | 39.96 | 2 | 138 | 6 | 4.26 | 56.3 | 134.5 | - | 0.1 |
| Palm | 870 | 34.4 L, 40.13 H | 14.25 | 176.7 | 14.33 | 4.53 | 60.21 | 50.5 | 2 | 0.2 |
| Peanut | 878 | 35.33 | 12.6 | 176 | 11.5 | 4.69 | 58.24 | 67.45 | 6 ppm | - |
| Plastic pyrolysis oil | 981 | 38.3 | - | 13 | - | 1.91 | - | - | 0.155 | 41 |
| Pont water algae | 872 | 40.8 | - | - | −16 | 5.82 | - | - | - | 0.4 |
| Poultry fat | 877 | 38.58 | - | 172 | 3 | 6.86 | - | 78.8 | - | 0.55 |
| Rapeseed | 879 | 35.8 L, 41.1 H | −3.5 | 169.5 | −11 | 4.4 | 48.25 | 112 | 0.0024 | 0.26 |
| Rice bran | 889 | 38.17 | 0.55 | 157.4 | 6.4 | 5.15 | 64.95 | 106 | 6 | - |

**Table 3.** *Cont.*

| Feedstocks | Density at 15 °C (kg/m³) | Heating Value (MJ/kg) | Cloud Point (°C) | Flashpoint (°C) | Pour Point (°C) | Viscosity at 40 °C (mm²/s) | Cetane Number | Iodine Number | Sulfur Content (wt.%) | Add Value mg/g |
|---|---|---|---|---|---|---|---|---|---|---|
| Rubber | 875 | 39.174 | 3.1 | 173.4 | −7 | 5.6 | 53 | 144 | - | 0.12 |
| Sesame | 867 | 40.25 | 0.5 | 176.67 | −4 | 4.23 | 58.97 | 83.52 | < 0.005 | 0.285 |
| Sludge pyrolysis oil | 980 | 37.04 | - | 68 | - | 12.3 | - | - | 0.55 | 26 |
| Sour plum | - | - | - | - | −6 | - | 61.39 | - | - | - |
| Soybean | 882 | 39.84 H | 0 | 140.1 | −3.2 | 4.15 | 44.7 | 117.7 | - | 0.18 |
| Spirulina | 860 | 41.36 | - | 130 | −18 | 5.66 | - | - | - | 0.45 |
| Spirulina platensis | 863 | 45.63 | −3 | 189 | −9 | 12.4 | 70 | 102 | 0 | - |
| Sunflower | 869 | 34.71 L, 40.6 H | 1.33 | 180.33 | −2 | 4.26 | 45.7 | 128.7 | - | 0.357 |
| Terminalia catappa | 876 | 37.33 | - | 90 | 6 | 4.3 | 57.1 | 83.2 | 13.3 | 0.5 |
| Tobacco | 865 | 42.22 | - | 165 | −12 | 3.56 | 51.5 | 136 | - | - |
| Trout oil | 885 | 37.8 | - | - | - | 4.25 | 51.3 | - | - | - |
| Waste cooking oil | 876 | 39.76 | - | 160 | - | 3.65 | 50.4 | 62 | - | - |
| Waste fry oil | 855 | 40.5 | −12 | 126 | - | 4.57 | 52.2 | - | - | - |
| Waste frying palm oil | 875 | 38.73 | - | 70.6 | - | 4.4 | 60.4 | - | - | 0.51 |

### 6.1. Cetane Number

The cetane number (CN) represents the ignition behavior and quality of the fuel. It affects the ignition delay time of the fuel. The ignition delay is the time gap between the fuel supply and the start of ignition. A high CN value indicates the quick self-ignition capacity of fuel [82]. The CN value of biodiesel increases with the length of the fatty acid chain and degree of saturation, so a higher CN means a higher oxygen concentration in the biodiesel and a better combustion efficiency [83]. The CN value of biodiesel is recommended by EN ISO 5165, ASTM D613, and ISO 5156/P9. The lowest value of the CN is 47 (ASTM standard), but in European and Indian standards, the minimum value is 51 [82]. Spirulina platensis microalgae-generated biodiesel showed the highest CN value of 70 [84]; however, biodiesel from linseed oil has the lowest CN value of 34.6 [85,86].

### 6.2. Cloud Point

The cloud point (CP) represents the minimum temperature at which the wax present in the fuel forms crystals, resulting in a cloudy appearance [83]. The CP fluctuates with the amount and nature of saturated fatty compounds. Biodiesel has higher CPs due to the high melting points of saturated fatty acids compared to unsaturated fatty acids. The standard procedure to determine the CP for biodiesel is explained in ASTM D2500 within a temperature range of $-3\,°C$ to $-12\,°C$. The lowest CP value for biodiesel produced from waste frying oil is $-12\,°C$ [87], whereas the highest CP value generated from mustard oil is $16\,°C$ [88].

### 6.3. Oxidative Stability

The oxidative stability denotes the resistance of the fuel against oxidation, and it is an essential factor that significantly influences the storage duration and condition [89]. Biodiesel containing a higher oxygen content is highly susceptible to oxidation deterioration. The degree of oxidation among biodiesels fluctuates according to their fatty acid composition [90]. The fuel oxidation stability is largely affected by polyunsaturated FAME. Camelina oil-based biodiesel showed a low oxidative stability, mainly due to 35% triunsaturated FAME occurrence, contrary to the coconut-oil-based biodiesel, which revealed a better oxidation stability due to 2% polyunsaturated FAME in its oil.

### 6.4. Viscosity

The viscosity indicates the flow capability of the fuel. Biodiesel has a higher viscosity, owing to a larger molecular mass and chemical structure than traditional fossil fuel [91,92]. A higher viscosity reduces thermal efficiency, whereas a lower viscosity facilitates easy fuel delivery in the combustion chamber [93]. The biodiesel kinematic viscosity is determined by following the ASTM D445, ISO 3104/P25, and EN ISO 3104 standards. The biodiesel kinematic viscosity ranges between 1.91–17.14 mm$^2$/s. In the case of biodiesel, spirulina platensis microalgae biodiesel fuel showed a maximum kinematic viscosity (17.14 mm$^2$/s), whereas a minimum viscosity (1.91 mm$^2$/s) was noted for plastic pyrolysis oil [81].

### 6.5. Lubricity

The lubricity specifies the reduction in friction force present between the two parts of a machine that have relative motion. Biodiesel fuels produced from different feedstocks hold good lubrication properties. They can be used to improve the lubricity of ultra-low sulfur diesel through mixing. Biodiesel (B100) contains a good lubricity due to the ester of FAME elements and impurities [81].

### 6.6. Density

The density is used to quantify the fuel amount supplied by the injector for the combustion process [83]. The fuel density affects the amount of energy and the air—fuel (A/F) ratio in the combustion chamber. The type of feedstock, methyl ester profile, and biodiesel production process influence the biodiesel density [94]. The biodiesel density can

be determined by following the test procedure of ASTM D1298, ISO 3675/P32, and EN ISO 3675/12185. The density of biodiesel belonging to various feedstock generations lies between 832–982 kg/m$^3$. Neem seed pyrolysis-oil-based biodiesel has the highest density (982 kg/m$^3$), whereas biodiesel from beef tallow oil has the lowest density (832 kg/m$^3$) [81].

### 6.7. Flashpoint

The flashpoint (FP) is the minimum temperature at which the fuel vapors catch fire when they come into contact with any ignition source. The flashpoint of biodiesel fuels is more than 150 °C, whereas common diesel has a flashpoint value of 55–65 °C. ASTM D93, P21, and EN ISO 3679 standards are used to describe the FP evaluation methods. The maximum FP value of 241 °C is reported for biodiesel generated from linseed, whereas the minimum FP value (55 °C) is noted in the biodiesel of neem seed pyrolysis oil [86,92,95].

### 6.8. Pour Point

The lowest temperature at which flow properties of liquid fuel are lost is known as the pour point (PP) [83]. The ASTM D97 methodology is followed to find out the PP value for biodiesel fuel. The minimum PP value of −25.1 °C is reported for castor oil, whereas the highest PP value of 15 °C is reported for mahua oil. Further, the minimum and maximum PP values of −20 °C and 15.5 °C are reported for castor-oil-based biodiesel and camelus dromedaries' fat-based biodiesel, respectively [96,97].

### 6.9. Cold Filter Plugging Point

The cold filter plugging point (CFPP) is the minimum temperature at which the discharge of the sample fuel takes place through a typical precise filter [83]. The CFPP represents the cold flow operation ability of the fuel and ASTM D6371 and EN 14,214 standards are employed to estimate the biodiesel CFPP. Tallow and palm-oil-derived biodiesel have a high temperature point, which means that they have a poor performance, whereas rapeseed-oil-based biodiesels have a low temperature point, so they show a good performance [81].

### 6.10. Calorific Value

The calorific value (CV) indicates the amount of energy released when a unit quantity of the fuel is burned. A fuel with a greater CV value is more beneficial for an internal combustion engine. Biodiesel fuels usually have a lower CV compared to petroleum diesel. Biodiesel derived from Spirulina platensis microalgae shows the maximum CV value of 45.63 MJ/kg, whereas neem seed pyrolysis-based biodiesel has the minimum CV value of 20.80 MJ/kg [81].

### 6.11. Acid Number

The acid number (AN) refers to the quantity of free fatty acids in the fuel sample. A higher acid number causes corrosion problems in the fuel delivery system of the engine [98]. The highest AN of 41 mg KOH/g is noticed in plastic pyrolysis-derived biodiesel, whereas the lowest AN of 0.1 mg KOH/g is reported for biodiesel from olive pomace [99,100].

### 6.12. Iodine Number

The iodine number (IN) indicates the quantity of iodine absorbed by double bonds of the FAME molecules in 100 g of the fuel sample. Linseed oil has the maxim IN value (156.74), whereas coconut oil has a minimum IN value of 10. In the case of biodiesel fuels, linseed methyl ester depicted the highest IN value of 178, whereas the lowest IN value of 37.59 is reported for kusum-oil-based biodiesel [86,98].

## 7. Methods of Biodiesel Production

Several technologies are currently used for biodiesel production. However, four significant methods are commonly used to modify crude oils to produce a substance that

has appropriate properties to be used as a fuel in engines. These methods include blending, micro-emulsion, thermal cracking, and transesterification [101,102].

### 7.1. Blending

Mixing pure vegetable oils with conventional fossil fuels to produce biodiesel is known as blending or dilution. A mixture of 20% vegetable oils and 80% diesel fuel can be used in compression ignition (CI) engines [103]. However, their direct use in diesel engines is problematic due to the high viscosity of the mixture; blending or mixing improves the viscosity of crude vegetable oils. A high viscosity causes more carbon deposits on the piston head, injector nozzle choking, and the deposition of gum. In compression ignition engines, a blend of 20% vegetable oil and 80% diesel fuel can be used directly [103,104].

### 7.2. Microemulsion Method

Micro-emulsification is a method of creating a stable colloidal dispersion of optically isotropic liquid mixtures (oil, water, and surfactant). Microemulsion resolves the high viscosity problem of vegetable oils. The microemulsion consists of a definite proportion of conventional diesel fuel, vegetable oil, surfactant, cetane improver, and alcohol. Microemulsions of hexanol, methanol, octanol, ethanol, and butanol solvents proved sufficient in meeting the required viscosity limit in diesel engines. However, some problems, such as carbon deposits, the stabbing of the injector nozzle, and incomplete combustion, may arise due to the continuous use of emulsified fuel [103,104].

### 7.3. Pyrolysis or Thermal Cracking

The process of converting raw oil into biodiesel fuel using heat or with the aid of heat and a catalyst is known as pyrolysis or thermal cracking. No oxygen or air is needed in the heating process and the temperature lies between 400–600 °C. The rate of pyrolysis controls the production of various products, such as char, biodiesel, and gases [101]. Depending on operational conditions, the pyrolysis process is categorized into conventional, flash, and fast pyrolysis. Properties of pyrolysis-based biodiesel are comparable to petroleum diesel [105]. Pyrolyzed vegetable oil has a lower viscosity and lower CN, flashpoint, and pour point values compared to diesel fuel. The equipment used for thermal cracking or pyrolysis is somewhat expensive [104]. Figure 2 represents the work flow of the pyrolysis process.

### 7.4. Transesterification

In the transesterification process, also known as alcoholysis [106], triglycerides and alcohols in the presence of a catalyst generate esters and glycerol. Transesterification involves three consecutive reversible reactions and, in each reaction, glycerides are converted into glycerol, and an ester is formed (Figure 3). The catalyst serves the purpose of accelerating the reaction rate in order to accomplish the process in minimum time. The major variables that affect the transesterification reactions include the temperature, reaction pressure, reaction time, oil to alcohol ratio, amount and type of catalyst, and feedstock type [107,108].

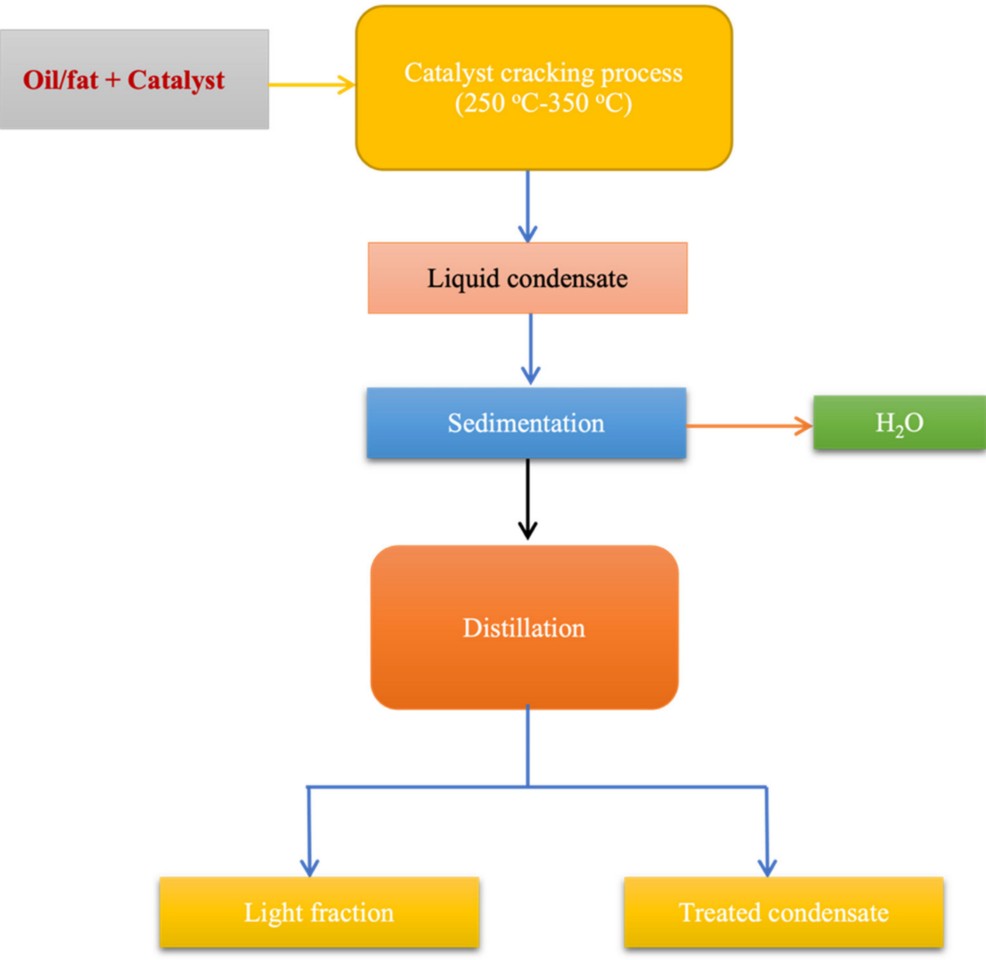

**Figure 2.** Flow chart representing the process of thermal cracking or pyrolysis.

### 7.4.1. Catalytic Transesterification

The transesterification (or alcoholysis) of vegetable oils is carried out by heating them with a catalyst and alcohol. Catalytic transesterification is of two kinds, homogeneous and heterogeneous, based on the type of catalyst used [104]. In homogeneous catalytic transesterification, the catalyst retains the same phase (liquid) as the reactants, whereas, in heterogeneous catalytic transesterification, the catalyst sustains a different phase (immiscible liquid, gaseous or solid) from the reactants. The homogeneous catalyst cannot be reused again, and it is difficult to separate glycerol and biodiesel products. These catalysts are corrosive and there is a high cost involved during the wastewater treatment system in neutralizing the catalyst before discharging. However, the heterogeneous catalyst can easily separate biodiesel and glycerol and can be reused, and there is no water or less water produced. The overall production cost of biodiesel could be reduced by using a heterogenous catalyst [109]. The selection of a suitable catalyst is highly important because it influences the cost of the biodiesel production [110].

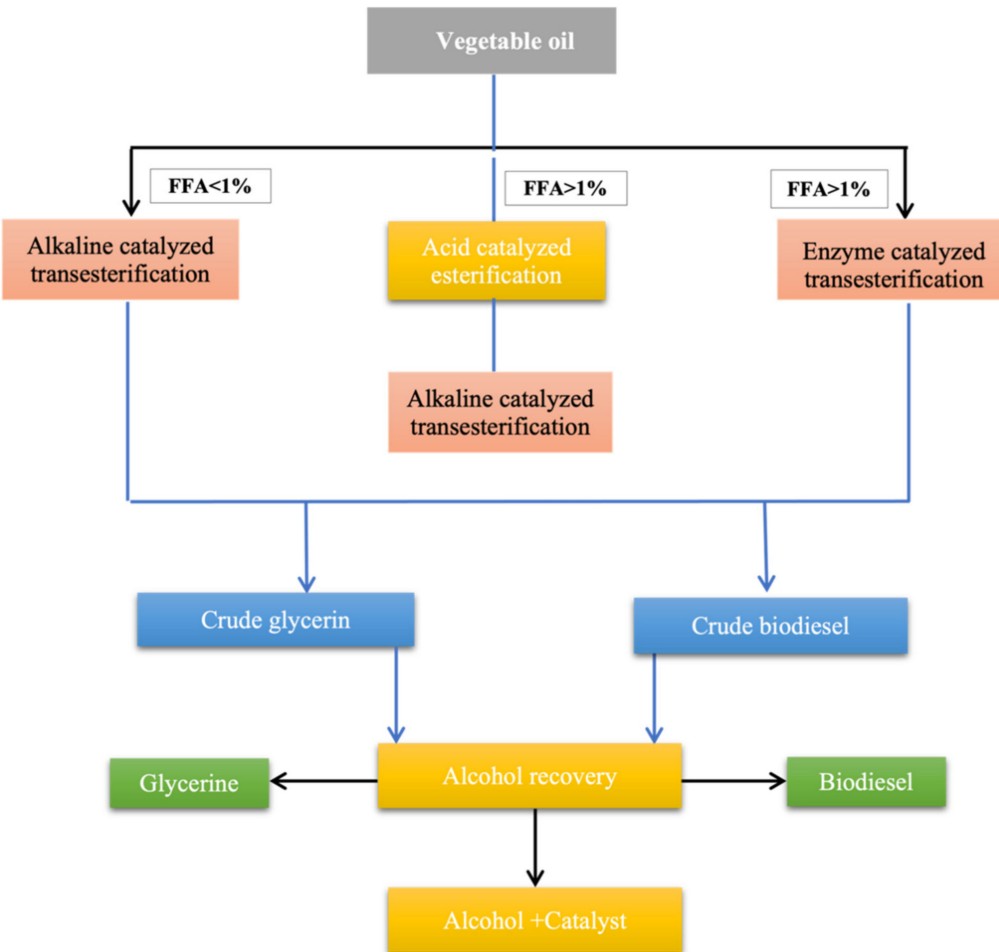

**Figure 3.** Biodiesel production through different transesterification methods.

### 7.4.2. Base or Alkaline Catalyzed Transesterification

This is the most common and popular method of biodiesel production [111]. Commonly used catalysts include alkaline metal alkoxides, sodium hydroxide (NaOH) or potassium hydroxide (KOH), and sodium methoxide (NaOCH$_3$). NaOH and KOH, with a concentration of 0.4% to 2% ($w/w$), are employed in the basic methanolysis process [112]. Using a NaOH homogenous catalyst, Jatropha oil is converted into methyl esters in 90 min and produces a 98% yield [113]. A low temperature (333–338 K), low pressure (1.4–4.2 bar), and catalytic concentrations (0.5–2 wt.%) are employed in this esterification [114]. However, it seeks additional purification requirements for biodiesel and glycerol for the removal of the basic catalyst. To avoid this drawback, researchers have also investigated heterogeneous catalysts for their potential to catalyze transesterification [115]. These catalysts include alkaline earth metal oxides, basic zeolites, and hydrotalcite. They are solid base catalysts that derive the heterogeneous catalytic process and ensure cost-effective biodiesel production. Heterogeneous catalysts can be easily separated from end products through a simple filtration process, and thus can be reused [104].

### 7.4.3. Acid-Catalyzed Transesterification

For oil containing a high amount of FFA, transesterification by acid catalysts (such as sulfuric acid, hydrochloric acid, and sulfonic acid) is most favored. However, alkaline-catalyzed transesterification is favored in oil with an FFA concentration of <1% [103].

In this method, oil is directly mixed with acidified alcohol to complete the separation and transesterification in one step, while alcohol acts as solvent as well as an esterification reagent. The excessive use of alcohol influences the reaction time needed for the homo-

geneous acid catalytic reaction. Therefore, the catalyst is required in large amounts for Brønsted acid catalytic transesterification to minimize the reaction time. Homogeneous acid-catalyzed transesterification is advantageous as it has a low susceptibility to FFA present in the feedstock. However, homogeneous acid-catalyzed transesterification is sensitive to water and the reaction is even completely abandoned if the water content exceeds 5% by weight. Additional problems include the corrosion of equipment, generation of secondary products, a long reaction time, higher reaction temperature, and weaker catalytic activity [116–119]. Acid-catalyzed esterification can provide a good foundation for base-catalyzed transesterification reactions by esterifying the FFAs if they are more than 2% [116,118,119].

### 7.4.4. Biocatalytic or Enzymatic Transesterification

Enzymes or biocatalysts are gaining popularity in catalyzing transesterification reactions compared to chemical (acid and alkali) catalysts [120]. Biocatalysts are naturally available lipases that have the potential to carry out transesterification reactions necessary for biodiesel production. Advantages of enzymatic transesterification include no by-product generation, easy product elimination, a low energy requirement for the process, higher biodiesel yield, and catalyst recycling [81]. Enzymatic reactions can be utilized for the transesterification of used cooking oil because they are insensitive to water and FFAs [121]. However, the application of enzymes in industries is expensive and the production of biodiesel through biocatalysts requires days in order to accomplish the transesterification reaction. Mass transfer limitation is the main cause of the slow processing of transesterification reactions [122].

### 7.4.5. Supercritical Alcohol Transesterification

This is a non-catalytic method in which the transesterification reaction is carried out at a high temperature and pressure in order to produce biodiesel [123]. It is a quick reaction and, in the starting 10 min, the conversion lies between 50–95% and needs a 250–400 °C temperature [124]. A relatively higher yield is obtained within a short time compared to the catalytic method. Moreover, the purification of biodiesel is easy and simple due to the absence of a catalyst and soap formation. The most promising transesterification of triglycerides is experienced with supercritical ethanol, methanol, butanol, and propanol. The requirements of a high temperature and pressure and higher alcohol to oil molar ratios make biodiesel production expensive and is a disadvantage of this method [104].

## 8. Jatropha as Second-Generation Biofuel Feedstocks from Non-Edible Source

### *8.1. Biodiesel Production Potential of J. curcas*

Due to the growing concern toward the competition for food, land, and water resources to produce first-generation bioenergy crops (for example, corn, sugarcane, and soybean), second-generation feedstocks of non-edible oil sources (such as Jatropha, mahua, caster) have gained much attention due to their widespread adaptability, even in adverse climatic and soil conditions of arid and semi-arid regions [2,9]. Further, they have a high oil content between 63.2–66.4%, which is much higher than that of soybean (18.6%), linseed (33.3%), and palm kernel (44.6%) [125]. Studies revealed that approximately 1.5–2.0 million ha of Jatropha have been planted in the last 5–7 years, resulting in around 13 million ha in 2015, and India is a leading country in terms of its cultivation (~73%), followed by Southeast Asia (21%) and Africa (6%) [2]. Jatropha oil can be used locally for diesel generators, cooking stoves, or fuel vehicles, without transesterification into biodiesel [126]. In developed countries, over 95% of biodiesel production feedstocks come from edible oils because the biodiesel produced from these oils has comparable properties to petroleum-based diesel [102]; however, vegetable oils have a great potential to substitute petroleum-based fuel in the long run [126]. Some of the species with biodiesel potential (i.e., non-edible oils) include mahua, karanja, caster, and linseed. The potential seed and biofuel yields for these species are summarized in Table 4.

**Table 4.** Production and oil content of non-edible oilseeds. This information is adapted from Ho et al. [2].

| Species | Seed Yield ($\times 10^5$ Mg ha$^{-1}$ Year$^{-1}$) | Oil Content (%) | Oil Yield (Mg ha$^{-1}$ Year$^{-1}$) |
|---|---|---|---|
| Jatropha | 2.0 | 40–60 | 2.0–3.0 |
| Mahua | 2.0 | 35–40 | 1.0–4.0 |
| Karanja | 0.6 | 30–40 | 2.0–4.0 |
| Caster | 2.5 | 45–60 | 0.5–1.0 |
| Linseed | 1.0 | 35–45 | 0.5–1.0 |

### 8.2. Fatty Acid Composition

The quantities of each fatty acid present in vegetable oils determine the properties of the triglyceride and biodiesel fuel [1,125]. There are three main types of fatty acids present in triglycerides, namely saturated (Cn:0), monounsaturated (Cn:1), and polyunsaturated with two or three bonds (Cn:2,3). Different vegetable oils are potential feedstocks that produce fatty acid methyl ester or biodiesel. However, the fuel quality is affected by the oil composition [1]. Vegetable oils from soybeans and sunflowers are rich in polyunsaturated fatty acid (PUFA), giving methyl ester fuels with poor oxidative stability (Table 4). Vegetable oils with a high degree of poly-unsaturation tend to have a high freezing point. This means that the oil has poor flow characteristics and might be easily solidified at low temperatures (e.g., palm oil), but would do well in hot climates [1].

The major long-chain fatty acids present in Jatropha seed oil are oleic, linoleic, palmitic, and stearic acid. Other fatty acids, such as myristic, palmitoleic, linolenic, arachidic, and behenatic, are also present, but in small amounts (Table 5). *J. curcas* seed oil displayed the highest composition of oleic acid, followed by linoleic acid [127]. Hence, Jatropha seed oil can be classified as oleic–linoleic oil (Table 5). Compared to the other vegetable oil, Jatropha oil has a higher oleic acid content than karanja, sunflower, soybean, and palm kernel oil (Table 5).

**Table 5.** Fatty acid composition (%) of different vegetable oils. This information is adapted from Koh and Ghazi [126].

| Fatty Acid | Structure | Formula | Composition (%) | | | | |
|---|---|---|---|---|---|---|---|
| | | | Jatropha Seed Oil [a] | Karanja Oil | Sunflower Oil | Soybean Oil | Palm Kernel Oil |
| Myristic | (14:0) | $C_{14}H_{28}O_2$ | 0–0.1 | - | - | 0.1 | 16.3 |
| Palmitic | (16:0) | $C_{16}H_{32}O_2$ | 14.1–15.3 | 9.8 | - | 11.0 | 8.4 |
| Palmitoleic | (16:1) | $C_{16}H_{16}O_2$ | 0–1.3 | - | - | - | - |
| Stearic | (18:0) | $C_{18}H_{36}O_2$ | 3.7–9.8 | 6.2 | 4.5 | 4.0 | 2.4 |
| Oleic | (18:1) | $C_{18}H_{34}O_2$ | 34.3–45.8 | 72.2 | 21.1 | 23.4 | 15.4 |
| Linoleic | (18:2) | $C_{18}H_{32}O_2$ | 29.0–44.2 | 11.8 | 66.2 | 53.2 | 2.4 |
| Linolenic | (18:3) | $C_{18}H_{32}O_2$ | 0–0.3 | - | - | 7.8 | - |
| Arachidic | (20:0) | $C_{20}H_{40}O_2$ | 0–0.3 | - | 0.3 | - | 0.1 |
| Behenatic | (22:0) | $C_{22}H_{44}O_2$ | 0–0.2 | - | - | - | - |

[a] [127].

### 8.3. Composition of J. curcas Oil

Jatropha oil contains a high level of triglycerides. The composition of triglycerides with C16–C18 corresponds to the hydrocarbon structure of diesel. Thus, Jatropha oil could be used directly as diesel as a road transportation fuel without further processing [128]. Generally, standard biodiesel and *J. curcas* oil compare well to petroleum-based diesel [5]. Jatropha biodiesel provides safety benefits over diesel fuel because it is much less combustible, with a flashpoint greater than that of diesel fuel. The CN value of Jatropha biodiesel is comparable to that of diesel fuel. Therefore, biodiesel from Jatropha oil would be effective as a substitute for diesel, while still retaining a higher cetane number [125]. The composition and fuel characteristics of Jatropha biodiesel and diesel are summarized in Table 6.

**Table 6.** Fuel characteristics of Jatropha biodiesel vs. diesel oil [5,55,125,126].

| Characteristics | Jatropha Biodiesel | Diesel Oil |
|---|---|---|
| Specific gravity 15 °C | 0.86–0.93 | 0.82–0.86 |
| Calorie value (MJ kg$^{-1}$) | 38–42 | 42 |
| Pour point (°C) | −3 | −35 to 15 |
| Cloud point (°C) | 2 | −15 to 5 |
| Flashpoint (°C) | 210–240 | 50–98 |
| Cetane number | 38–51, up to 57 | 40–55 |
| Sulfur | 0.13 | 1.2 |
| Viscosity (cSt) at 30 °C | 37–55 | 1.3–4.1 |

## 9. Environmental Impacts and Economic Aspects of Jatropha Cultivation

Producing and using biodiesel from biofuel crops, such as Jatropha, for transportation provides alternatives to fossil fuels that help to resolve environmental problems [8]. Emissions from biofuels are much lower compared to emissions from conventional fuels due to the low or zero content of pollutants including, sulfur, in biofuels (e.g., $SO_2$). The use of biofuels in motor vehicles can help to reduce the burning of fossil fuels and reduce the amount of greenhouse gas (GHG) emissions [93]. Studies revealed that biodiesel produced from *J. curcas* reduces both greenhouse gas (GHG) emissions by ~8–88% and the nonrenewable energy demand compared to fossil-based diesel [129]. However, a life-cycle assessment of *Jatropha* in Mexico showed that the use of Jatropha biodiesel from plantations on uncultivated lands (with high carbon) can increase the GHG emissions by three to six times compared to the burning of fossil fuels [130]. To balance the GHG emissions from Jatropha cultivation, its seed yield must be increased from a current yield of <1700 kg ha$^{-1}$ to >4800 kg ha$^{-1}$ [130]. Another best alternative to reduce GHG emissions, particularly $CO_2$, is by using a blend of 20% biodiesel, which lowers $CO_2$ emission by 15%. Previous studies reported that engine operations using biodiesel mixed with diesel produced comparatively lower emissions than diesel fuel, except for NOx emissions, where there is a 2% and 10% increase in emissions with B20 and B100 blending, respectively [55]. Overall, the use of biodiesel can reduce the burning of fossil fuels and help to save carbon reserves. However, negative environmental impacts associated with fossil fuels are acidification, eutrophication, ecotoxicity, water depletion, and deforestation, which could easily be reduced by the efficient use of fertilizers and site-specific management during Jatropha cultivation [129].

Major costs associated with Jatropha cultivation include establishment costs (land purchase/lease, equipment, farm preparation, seedling, and planting), management costs (insect pest, disease, irrigation, and nutrient management), and harvest and post-harvest costs (harvest, peeling, seed preparation, transportation, and marketing). However, due to variations in factors associated with the manpower requirement and wage rate in various geographical locations, the economic viability of its cultivation solely depends on the input costs, seed yield, and market price [131]. Several studies (i.e., 37) conducted from 26 countries across the world evaluated the economic feasibility of Jatropha cultivation and found comparative results (i.e., 10 were positive, 11 were negative, and 16 showed neutral results). They concluded that the economic viability can be obtained for small-scale farming if costs associated with labor and lands are very low. Jatropha cultivation is profitable if seed yields are > 2000–2500 kg ha$^{-1}$ year$^{-1}$, with reliable markets for selling its by-products, such as seed cake and glycerine [132].

A study conducted in Rwanda using the net present value (NPV) and benefit–cost ratio (BCR) revealed that Jatropha cultivation is not economically viable, due to high production costs, low seed yields, and a low Jatropha seed price. However, a sensitivity analysis found it to be sustainable when the Jatropha seed yield is >7000 kg ha$^{-1}$ year$^{-1}$. Further, the sensitivity analysis also suggested that small-scale Jatropha cultivation can be economically viable when NPV = $450 ha$^{-1}$ and BCR = 4.16 when the loan is not charged interest [131]. A similar study conducted in Northeast India displayed positive returns on Jatropha plantations, making it an economically feasible venture for the farmers of that

region. However, it showed a payback period of 5 years on investment when seed yields were higher and the government funding support for the operation and management of plantations during the initial years was adequate [133]. Additionally, studies found no economic viability from Jatropha biodiesel in Nepal unless seed yields are very high (>5000 kg ha$^{-1}$ year$^{-1}$) with a carbon credit of US$50/tCO$_2$, and when there are high economic returns from by-product markets, such as glycerol and Jatropha cake [134].

## 10. Conclusions

This study has provided a comprehensive review of *J. curcas*, a second-generation biofuel crop for sustainable biodiesel production in marginal lands of arid and semi-arid regions of the world. It focused on every aspect of the crop, starting from its origin and distribution, genetic diversity, agronomic aspects, its productivity, oil content, and fatty acid profile, methods for extracting biodiesel production, as well as economic and environmental considerations for biodiesel production. According to literature, it is suggested that biodiesel production from *J. curcas* can reduce greenhouse gas emissions by 8-88% compared to fossil-based diesel. With a higher seed yield production, government funding support, a payback period of at least 5 years, and the use of Jatropha by-products, such as glycerol, from countries including Rwanda, Nepal and India can be highly benefited from Jatropha production. Although the crop has a high potential to be a biofuel crop, there are still knowledge gaps that need to be considered in order to optimize the oil production from this crop, such as an investigation of the taxonomic status of existing plantations, the agronomic potential of other *Jatropha* spp., such as *J. canescens*, the initiation of a breeding program with multilocation testing of promising provenances, an economic analysis of seed oil production in rural areas for biodiesel fuel and soap production, and socioeconomic studies on how *Jatropha* spp., can aid development in local communities.

Based on various literatures reviewed, it can be concluded that there is a wide research opportunity in terms of the production process, improving the performance, and emission reduction for biodiesels using Jatropha. There is also an enormous research opportunity in the area of biodiesel yield improvement. Further, genome engineering, such as metabolic engineering, can be implemented in order to increase the yield of the biodiesel production. The production cost for the biodiesel from Jatropha is very high; hence, there is a need to optimize the cost of biodiesel production.

**Author Contributions:** Conceptualization, D.N.; writing—original draft preparation, D.N., D.B., Z.A., S.P., and P.A.; writing—review and editing, D.N., D.B., Z.A., B.D., S.P., J.K.Q.S., R.Q., and P.A.; compilation and supervision, D.N. All authors have read and agreed to the published version of the manuscript.

**Funding:** There is no external funding for this publication.

**Data Availability Statement:** There is not any primary data used in this study, however, secondary data that supports of this study are available from the corresponding author (D.N.) upon reasonable request.

**Acknowledgments:** The authors would like to thank Nick Howard from 6 Engineering and an associate member of the IChemE, UK for his critical review of the article. We also would like to thank the anonymous reviewers and editors for their valuable comments and suggestions to improve the quality of this article.

**Conflicts of Interest:** The authors declare no conflict of interest or personal relationship that could have appeared to influence the work reported in this study.

## Abbreviations

| | |
|---|---|
| J. curcas | Jatropha curcas L. |
| EN ISO | European Standards International Organization for Standardization |
| ASTM | American Society for Testing and Materials |
| FAME | fatty acid methyl ester |

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
