# Peer review of "Growing Jatropha (Jatropha curcas L.) as a Potential Second-Generation Biodiesel Feedstock"

_inventions, doi:10.3390/inventions6040060_

Round 1

Reviewer 1 Report

The article is very informative and very well presented. In my opinion, it should be accepted after minor changes:

Table 2: How has the table been adapted? I imagine each row of the table refers to one study, so I believe each reference should appear at the final column, so that it is accessible in case a reader wants to access to that information. There are some repeated feedstocks in the table: please, try to assemble the information for an easier reading.

Please, also add the references to the rest of tables.

In my opinion, Section 8 should go after Section 4

Author Response

Comments and Suggestions for Authors

The article is very informative and very well presented. In my opinion, it should be accepted after minor changes.

Response: Thank you very much for your valuable comments and suggestions which definitely improve the overall quality of our manuscript.

Comments: Table 2: How has the table been adapted? I imagine each row of the table refers to one study, so I believe each reference should appear in the final column so that it is accessible in case a reader wants to access that information. There are some repeated feedstocks in the table: please, try to assemble the information for easier reading.

Please, also add the references to the rest of the tables.

Response: Table 2 has been adapted from Singh et al.,[80]. https://doi.org/10.1016/j.fuel.2019.04.174.

 Singh et al., 2019 cited articles for individual feedstock. However, we decided not to cite for the individual feedstock thinking that the table might look more crowded with the additional columns, also other articles use the same format.

Comments: In my opinion, Section 8 should go after Section 4

Response: We moved to section 8 just after section 4 as per the reviewer’s suggestion, which looks great.

Reviewer 2 Report

Inventions 1389154

 Growing Jatropha (Jatropha curcus L.) as a Potential 2nd Generation Biodiesel Feedstock

Authors: Dhurba Neupane, Dwarika Bhattarai, Zeeshan Ahmed, Bhupendra Das, Sharad Pandey, Juan K.Q.  Solomon, Ruijun Qin and Pramila Adhikari

The article is a review about the use of Jatropha as a potential feedstock for biodiesel

The article is well written easy to follow and it can be accepted after minor revision

Minor points

Please read carefully the manuscript for typos syntax errors etc. For example only in abstract

 iden-tified,  threaten-ing etc

Figure 1 should be provided in higher analysis

Line 156 please correct with Singh, et al instead of all the authors names

Table 2 the units of density should be added

In transesterification part the use of solid catalysts should be also shortly discussed.

The review Catalysts 2018, 8(11), 562; https://doi.org/10.3390/catal8110562 can be added in references

The part 9 can be further discussed and more information can be added

Author Response

Comments: The article is well written easy to follow and it can be accepted after minor revision

Response: Thank you for your kind and priceless comments and suggestions.

Comments: Please read carefully the manuscript for typos syntax errors etc. For example, only in the abstract

 iden-tified, threaten-ing etc.

Response: We have corrected the typos syntax errors in the abstract and other places as per your suggestions.

 Comments: Figure 1 should be provided in higher analysis.

Response: Figure 1 is updated with better resolution.

Comments: Line 156 please correct with Singh, et al instead of all the author's names.

Response: In Line 156, the reference is corrected.

Comments: Table 2 the units of density that should be added.

Response: The unit of density is added.

Comments: In the transesterification part, the use of solid catalysts should be also shortly discussed.

The review Catalysts 2018, 8(11), 562; https://doi.org/10.3390/catal8110562 can be added in references

Response: We have added a couple of sentences about the use of solid catalysts. The suggested reference has been added.

Comments: The part 9 can be further discussed and more information can be added

Response: We think section 9 is already long enough, and there is sufficient information regarding the section. The article is already long, so we do not want to add further detail and make it lengthy. Sorry for not elaborating more information on it.

Reviewer 3 Report

The subject of the article is very interesting and may turn out to be an interesting and profitable solution in the future.

Please consider whether it is necessary to refer to the same source more than once in the same paragraph, e.g.:

  • "Further, the use of vegetable-based products helps to promote rural economic development, particularly in developing countries, because the farmers would directly benefit from increased demand for vegetable oils [2]. This will help to decrease dependency on fossil fuel imports, improve their economic conditions, and can create new employment opportunities, especially in the agriculture area [7]. Vegetable oils, such as palm oil, soybean oil, sunflower oil, canola oil, and rapeseed oil have been used to generate biodiesel fuel [2]. The use of biodiesel as a fuel has been increased substantially in recent years, however, feedstock cost accounts for a greater percentage of direct biodiesel production cost, including capital cost and return [8]. Moreover, the use of first-generation biofuel crops such as corn, sugarcane, wheat, sugar beet, cassava, rapeseed, soybean, and oil palm, which contribute to more than two thirds of the bioenergy demand of the world, pose a growing concern over competition for land, food and water resources for the production of energy [1,9,10]. One of the potential approaches to reducing biodiesel production cost is to use low-cost feedstock containing fatty acids, such as inedible oils, animal fats, by-products from refining vegetable oils, and waste food oils [2]."
  • "However, negative environmental impacts such as acidification, eutrophication, ecotoxicity and water depletion [127], and even deforestation [129] should be considered. These negative impacts could easily be reduced by the efficient use of fertilizers and site selection during Jatropha cultivation [127]."

The Conclusion chapter must be completed and corrected. For example, please indicate specific solutions that could lead to greater interest in this plant (Jatropha). Please also emphasize the importance of the analyzes carried out.

Please adapt the manuscript to the journal's requirements, eg in References, some article titles are bold and should be italicized. 

Author Response

Comments: The subject of the article is very interesting and may turn out to be an interesting and profitable solution in the future.

Response: Again, we would like to thank the reviewer for his/her valuable comments and suggestions.

Comments: Please consider whether it is necessary to refer to the same source more than once in the same paragraph, e.g.:

  • "Further, the use of vegetable-based products helps to promote rural economic development, particularly in developing countries, because the farmers would directly benefit from increased demand for vegetable oils [2]. This will help to decrease dependency on fossil fuel imports, improve their economic conditions, and can create new employment opportunities, especially in the agriculture area [7]. Vegetable oils, such as palm oil, soybean oil, sunflower oil, canola oil, and rapeseed oil have been used to generate biodiesel fuel [2]. The use of biodiesel as a fuel has been increased substantially in recent years, however, feedstock cost accounts for a greater percentage of direct biodiesel production cost, including capital cost and return [8]. Moreover, the use of first-generation biofuel crops such as corn, sugarcane, wheat, sugar beet, cassava, rapeseed, soybean, and oil palm, which contribute to more than two thirds of the bioenergy demand of the world, pose a growing concern over competition for land, food and water resources for the production of energy [1,9,10]. One of the potential approaches to reducing biodiesel production cost is to use low-cost feedstock containing fatty acids, such as inedible oils, animal fats, by-products from refining vegetable oils, and waste food oils [2]."
  • "However, negative environmental impacts such as acidification, eutrophication, ecotoxicity and water depletion [127], and even deforestation [129] should be considered. These negative impacts could easily be reduced by the efficient use of fertilizers and site selection during Jatropha cultivation [127]."

Response: We have considered the comments regarding the use of the same source used more than once in the same paragraph. We tried our best to address the points. In most of the cases, we combined information and cited only one time

Comments: The Conclusion chapter must be completed and corrected. For example, please indicate specific solutions that could lead to greater interest in this plant (Jatropha). Please also emphasize the importance of the analyzes carried out.

Response: The conclusion part is completed and corrected as per your suggestion. We provided specific solutions that could lead to greater interest in this crop.

Comments: Please adapt the manuscript to the journal's requirements, eg in References, some article titles are bold and should be italicized. 

Response: We corrected some of the issues that appeared in the references, and strongly followed the journal’s requirements.

Thank you very much!
